# Identification of Molecular Subtypes and Prognostic Characteristics of Adrenocortical Carcinoma Based on Unsupervised Clustering

**DOI:** 10.3390/ijms242015465

**Published:** 2023-10-23

**Authors:** Yuan Zhang, Cong Zhang, Kangjie Li, Jielian Deng, Hui Liu, Guichuan Lai, Biao Xie, Xiaoni Zhong

**Affiliations:** Department of Epidemiology and Health Statistics, School of Public Health, Chongqing Medical University, Yixue Road, Chongqing 400016, China; 2022110598@stu.cqmu.edu.cn (Y.Z.); 2022120781@stu.cqmu.edu.cn (C.Z.); 2021160020@stu.cqmu.edu.cn (K.L.); 2021120746@stu.cqmu.edu.cn (J.D.); 2020121415@stu.cqmu.edu.cn (H.L.); 2020111425@stu.cqmu.edu.cn (G.L.)

**Keywords:** adrenocortical carcinoma, unsupervised clustering, immune-related genes, subtype, biomarkers, prognosis

## Abstract

Adrenocortical carcinoma (ACC) is a rare endocrine malignancy with a poor prognosis. Increasing evidence highlights the significant role of immune-related genes (IRGs) in ACC progression and immunotherapy, but the research is still limited. Based on the Cancer Genome Atlas (TCGA) database, immune-related molecular subtypes were identified by unsupervised consensus clustering. Univariate Cox analysis and Least Absolute Shrinkage and Selection Operator (LASSO) regression were employed to further establish immune-related gene signatures (IRGS). An evaluation of immune cell infiltration, biological function, tumor mutation burden (TMB), predicted immunotherapy response, and drug sensitivity in ACC patients was conducted to elucidate the applicative efficacy of IRGS in precision therapy. ACC patients were divided into two molecular subtypes through consistent clustering. Furthermore, the 3-gene signature (including PRKCA, LTBP1, and BIRC5) based on two molecular subtypes demonstrated consistent prognostic efficacy across the TCGA and GEO datasets and emerged as an independent prognostic factor. The low-risk group exhibited heightened immune cell infiltration, TMB, and immune checkpoint inhibitors (ICIs), associated with a favorable prognosis. Pathways associated with drug metabolism, hormone regulation, and metabolism were activated in the low-risk group. In conclusion, our findings suggest IRGS can be used as an independent prognostic biomarker, providing a foundation for shaping future ACC immunotherapy strategies.

## 1. Introduction

Adrenocortical carcinoma (ACC) is a rare and aggressive tumor that starts in the adrenal cortex. It is worth mentioning that approximately 40% to 60% of patients show symptoms and signs that suggest an excessive production of adrenal steroids [1,2]. According to reports, the annual incidence of ACC is approximately 0.7–2 cases/million people, and the 5-year average survival rate is 20–25%. Nevertheless, the percentage of patients who survive for five years in stage IV is only 6–13% [3]. Currently, the sole treatment choice for individuals with non-metastatic ACC is complete removal of the tumor [4]. Continued exploration of ACC’s pathogenesis remains crucial. The existing comprehension of the disease’s mechanisms is rather unclear. The lack of clarity poses challenges for newly diagnosed ACC patients when assessing their risk levels in their initial appointments [5]. Hence, conducting thorough investigations to identify novel predictive biomarkers can not only provide deeper insights into the development of ACC metastasis but also facilitate the identification of fresh treatment targets [6,7].

Immune checkpoint inhibitor (ICIs) treatment has been an effective therapeutic tool that has dramatically changed the treatment pattern of solid tumors, including ACC, head and neck squamous cell carcinoma (HNSCC), and colorectal cancer (CRC) [8,9,10]. Previous studies have found that patients’ age, tumor grade, and tumor range may have an impact on the prognosis of ACC [11,12,13]. However, the range of predictive markers has broadened in recent studies. The response of patients treated with ICIs may be influenced by immune-related genes (IRGs), such as programmed death-1 (PD-1) and PD-L1 in the tumor microenvironment (TME) [14,15], tumor-infiltrating immune cells (TICs) [16,17], tumor mutation burden (TMB) [18], and the expression of immune-related signaling pathways [19] in the TME, which may have an impact on the clinical outcome of patients treated with ICIs [20]. Zhang et al. pointed out that the presence of a V domain Ig T cell activation inhibitor (VISTA) in tumor cells is linked to the advancement of ACC and an elevated overall mortality rate. The findings provide new evidence supporting the potential of VISTA as a viable target for immunotherapy in the treatment of ACC [21]. Interfering with the binding between PD-1 and PD-L1 has the potential to enhance T cell proliferation, release cytokines, eliminate infected cells, lower the viral load, and potentially boost the immune response against tumors [22]. According to certain research, the capacity of PD-L1 expression in both tumor cells and immune cells infiltrating tumors to serve as indicators of the response to immunomodulatory agents has been well documented [23]. Furthermore, the activation of Common γ-receptor-dependent cytokines and their JAK/STAT pathways is commonly observed in the majority of T-cell malignancies, potentially leading to a complete transformation in the approach to treating such malignancies [24]. In addition, Peng et al. discovered a pair of IRGs that exert a significant influence on tumor progression and the body’s immune response. These findings provide valuable information about the tumor microenvironment in ACC and indicate the possible usefulness of these genes as markers for immunotherapy in ACC [25]. The changing environment implies that the clinical results of patients receiving immunotherapy may be influenced not only by conventional demographic and histopathological factors but also by complex immune factors [26]. Hence, it is crucial to examine and assess the connection between the aforementioned factors and ACC immunotherapy, aiming to offer fresh perspectives for enhancing clinical diagnosis, prevention, and the advancement of novel treatment approaches [27].

In this work, using the unsupervised clustering method, we first classified ACC patients into distinct molecular subtypes according to IRGs. Additionally, we developed a three-gene signature using IRGs that were differentially expressed, enabling accurate evaluation of the prognostic risk in ACC patients. To investigate the potential significance of the three-gene signature and gain fresh perspectives on the disease’s pathogenesis and potential immunotherapy, we delved deeper into the correlation between gene signature and TICs, TMB, functional enrichment, drug sensitivity, and predictive immunotherapy. 

## 2. Results

### 2.1. Identification of Characteristics and Biological Functions of Molecular Subtypes

A grand total of 2483 IRGs were acquired from the Immunology Database and Analysis Portal (ImmPort) database. Based on the expression patterns of 2483 IRGs in TCGA, we conducted consensus unsupervised clustering on 79 samples from TCGA-ACC in order to investigate novel molecular classifications for ACC patients. Assess the clustering values (k) for a range of 2 to 7 outcomes. The CDF Delta plot shows that the clustering outcomes exhibit greater stability when k = 2, and the PAC algorithm and a consistent heatmap determine that the optimal number of subtypes is 2 (Figure 1A–C; Appendix A). In the comparison between patients of subtype 2 and subtype 1, subtype 2 patients exhibited significantly longer overall survival (OS), as indicated by the Kaplan–Meier (KM) survival curves (*p* = 0.0025, Figure 1D). To identify important IRGs in ACC, we utilized TCGA-ACC data to screen differentially expressed genes (DEGs) between subtypes 1 and 2. Additionally, 340 DEGs were identified based on a significance threshold of *p*-values < 0.05. Out of these, a total of 202 genes exhibited up-regulation while 138 genes showed down-regulation (Figure 1E and Appendix A). Through the examination of the Gene Ontology (GO) function of these immune-related DEGs, it was discovered that, in terms of biological processes (BP), they were primarily concentrated in cytokine-mediated signaling pathways and the regulation of leukocyte proliferation. Regarding cellular components (CC), the DEGs were predominantly engaged in the biological processes occurring on the external side of the plasma membrane and the MHC protein complex. Furthermore, molecular function (MF) exhibited activation in cytokine-mediated signaling pathways and receptor ligand activity (Figure 1F; Appendix A). Kyoto Encyclopedia of Genes and Genomes (KEGG) analysis showed that a total of 325 differential pathways were identified, of which the ten most significant pathways were Neuroactive ligand-receptor interaction, Hematopoietic cell lineage, Cytokine-cytokine receptor interaction, Cell adhesion molecules, Calcium signaling pathway, Graft-versus-host disease, Type I diabetes mellitus, Allograft rejection, Viral protein interaction with cytokine and cytokine receptor, PI3K-Akt signaling pathway (Figure 1G; Appendix A). The results indicate that DEGs are linked to functions and pathways related to the immune system in ACC.

### 2.2. Construction and Verification of IRGs Gene Signature

At present, 340 DEGs have been identified in TCGA-ACC samples. To further decrease the feature dimension, we acquired 120 genes associated with OS (*p* < 0.05) that were identified as prognostic genes in ACC patients by univariate Cox regression analysis. Afterwards, the 120 genes were incorporated into the Least Absolute Shrinkage and Selection Operator (LASSO) regression analysis, which was subsequently narrowed down to only three genes. In conclusion, a prognostic model (immune-related gene signature, IRGS) for ACC patients was established using three genes related to the immune system (PRKCA, LTBP1, and BIRC5). We then computed risk scores for each patient with ACC in both the training set and validation cohorts, using the subsequent mathematical equation: Riskscore = −0.00487 × PRKCA + 0.18795 × LTBP1 + 0.30888 × BIRC5. Based on the median risk score, the patients were categorized into high-risk and low-risk groups. The prognostic effectiveness of IRGS was evaluated using time-dependent receiver operating characteristic (ROC) curves and KM curves. The survival rate in the training set showed a notable disparity between the high-risk and low-risk groups (*p* < 0.001). In addition, patients in the high-risk group demonstrated a significant decrease in their overall survival duration (Figure 2A). In the TCGA-ACC, the area under the curve (AUC) values for 1-year, 3-year, and 5-year OS were 0.851, 0.954, and 0.898, respectively, indicating that IRGS has excellent precision in predicting survival duration for both high-risk and low-risk groups (Figure 2E). The correlation between the risk score and the duration and status of survival is shown in Figure 2D.

To confirm the effectiveness of the IRGS, it was validated in the GSE33371 and GSE10927 datasets. Consistent with the results observed in the TCGA-ACC training cohort, the high-risk group demonstrated a notably reduced survival rate in the validation cohorts compared to the low-risk group (GSE33371: *p* = 0.0028, GSE10927: *p* = 0.0048; Figure 2B,C). Moreover, we confirmed the robust performance of the ROC curve during our validation process (Appendix A). To thoroughly assess the model’s ability to differentiate, we generated a risk score and survival time distribution map for the GSE33371 and GSE10927 cohorts (Appendix A). To summarize, these findings indicate the strength of IRGS in predicting outcomes for ACC patients. 

### 2.3. IRGS and Prognostic Analysis of Clinical Characteristics

Next, we examined the association between the risk scores and different clinical attributes of the individuals. Figure 3A illustrates the utilization of a Sankey diagram to display the allocation of individuals across various risk categories. The Wilcoxon test results indicated significant variations in risk scores among various T stages, M stages, and clinical stages (Appendix A). Nevertheless, there was no variation in the average risk scores across gender, age, and N stage. Next, we investigated the significance of IRGS in biological processes. The high-risk and low-risk groups underwent GO enrichment and KEGG pathway analysis. The results of GO enrichment included the cytokine-mediated signaling pathway, cell chemotaxis, receptor ligand activity, signaling receptor activator activity, the external side of the plasma membrane, and the secretory granule lumen (Figure 3B; Appendix A). Furthermore, a grand total of 218 distinct KEGG pathways were discovered when comparing the high-risk and low-risk groups. Among these, the top ten pathways of utmost importance included cytokine-cytokine receptor interaction, viral protein interaction with cytokine and cytokine receptor, neuroactive ligand-receptor interaction, hematopoietic cell lineage, chemokine signaling pathway, IL-17 signaling pathway, rheumatoid arthritis, natural killer cell mediated cytotoxicity, NF-kappa B signaling pathway, and graft-versus-host disease (Figure 3C; Appendix A).

### 2.4. Independent Prognostic Indicator of IRGS 

In order to determine if the IRGS can function as an independent prognostic indicator, we incorporated various clinical factors, including age, gender, T stage, M stage, N stage, and clinical stage, into the evaluation of the IRGS in TCGA-ACC patients. Finally, prognosis was found to be associated with T stage, M stage, clinical stage, and risk score in the univariate Cox analysis (Figure 4A). Moreover, in the multivariate Cox analysis, the risk score was identified as a separate prognostic indicator for ACC patients (hazard ratio: 0.011–0.261, *p* < 0.001, Figure 4B). These findings were consistently validated in the GEO cohort (Appendix A). The aforementioned findings suggest that the IRGS holds promise as a reliable and independent prognostic marker for individuals with ACC.

### 2.5. Analysis of Tumor Immune Infiltration

We applied multiple algorithms to investigate variations in the level of immune infiltration in TCGA-ACC cohorts. According to our findings, the group with low risk showed decreased levels of tumor infiltration but displayed elevated immune and stromal scores, along with composite scores. In contrast, the high-risk group displayed a contrasting trend, characterized by increased levels of tumor infiltration and reduced immune and stromal scores, along with composite scores. We utilized the CIBERSOR algorithm to determine the infiltration percentage of 22 immune cells. In the high-risk group, there was a notable increase in the presence of activated dendritic cells, and macrophages M0. Conversely, the low-risk group exhibited elevated levels of M2 macrophages, activated NK cells, and CD8^+^T cells (Figure 5A). The proportion of infiltration by 64 immune cells was calculated using the xCell algorithm. The low-risk group had significantly higher immunity scores, stromal scores, and microenvironmental scores compared to the high-risk group. The risk score (Figure 5B–D) showed a negative correlation with the immune score, stromal score, and microenvironmental score. Afterwards, the ESTIMATE algorithm was employed to assess the tumor’s purity. Typically, when there is a higher presence of immune cells and stromal cells, the tumor’s purity tends to decrease. During this investigation, a comparison between the high-risk and low-risk groups revealed that patients in the latter group displayed improved immune scores and stromal scores (Appendix A). Furthermore, a significant inverse correlation was found between tumor purity and risk score (r = −0.31, *p* = 0.0061; Figure 5E–F). To summarize, these results indicate that the tumor’s ability to suppress the immune system could be a contributing factor to the unfavorable prognosis of ACC patients in the high-risk group.

### 2.6. The Correlation between IRGS and TMB

To assess the correlation between IRGS and TMB, we utilized the R package “maftools”. The genes MUC16, TP53, CTNNB1, TTN, ASXL3, CNTNAP5, PCDH15, PKHD1, DST, and HMCN1 were found to have the highest mutation frequencies in patients with ACC (Figure 6A). Upon computation of the TMB for patients with ACC, it was observed that the TCGA-ACC cohort had a median TMB of 0.42/MB (Figure 6B), leading to the categorization of patients into high-TMB and low-TMB groups. To establish the inherent relationship between TMB and IRGS, we conducted a comparison of the TMB variation in different risk groups. The results of our study showed a markedly elevated TMB in the high-risk cohort, which demonstrated a direct association with the risk score of patients with ACC (Figure 6C,F). Moreover, the low-TMB group showed a tendency towards improved OS (Figure 6D). The KM curve indicated that TMB status had no impact on the prediction of IRGS, and consistently, the low-risk group demonstrated a superior survival benefit (Figure 6E).

### 2.7. GSVA Analysis of KEGG and Immunologic Pathways between Different Risk Groups

By analyzing the enrichment score (ES) of 186 KEGG and 4872 immune gene sets, we discovered a combined count of 69 distinct KEGG pathways and 2385 immune pathways among the two risk groups. The low-risk group showed significant enrichment in tyrosine metabolism, steroid hormone biosynthesis, drug metabolism by cytochrome P450, metabolism of xenobiotics by cytochrome P450, aldosterone-regulated sodium reabsorption, primary bile acid bilsynthesis, the calcium signaling pathway and so on among the top 20 pathways of KEGG (Figure 7A; Appendix A). Meanwhile, we computed the enrichment score (ES) for 2385 gene sets associated with the immune system using Gene Set Variation Analysis (GSVA). The gene sets belonging to the high-risk group were markedly up-regulated in the top 20 gene sets, whereas the gene sets of the low-risk group exhibited significant down-regulation (Figure 7B, Appendix A).

### 2.8. Correlation of IRGS with Immunotherapy Indicators

According to the Wilcox test analysis, several immune checkpoints, such as CD8A, exhibit high expression in the low-risk group in immunotherapy (Figure 7C). To evaluate the predictive capability of IRGS to predict the response to ACC immunotherapy, the Immunophenoscore (IPS) algorithm was employed. The findings indicated a significant increase in the IPS score of the low-risk group, and a negative correlation was found between the risk score and the IPS score (Figure 8A,D). This implies that within the low-risk category, there could be a greater variety of immunophenotypes. Furthermore, patients classified as low-risk exhibited an elevated T cell-inflamed gene expression profile (GEP) score (Figure 8B,E) and MHC I association immunoscore (MAIS) score compared to those in the high-risk group (Figure 8C,F). The observation indicated that individuals in the low-risk category might exhibit a more positive reaction to immunotherapy.

### 2.9. Drug Sensitivity Analysis

Given the fact that certain individuals display resistance towards traditional medications, we investigated how patients with ACC respond to three frequently prescribed chemotherapy drugs. Notably, patients classified as high-risk may derive greater benefits from these commonly prescribed medications, as evidenced by the higher IC50 found in patients from the low-risk group. This finding provides valuable insights for guiding clinical treatment strategies (Figure 8G–I).

## 3. Discussion

The prognosis for ACC varies greatly. Some tumors can be cured by complete surgery, while others cannot be removed, and they grow rapidly. The possibility of metastasis and diffusion is great, resulting in a poor prognosis [28]. Despite the utilization of various clinical biomarkers in the assessment of ACC prognosis [29,30,31], the adequacy of risk stratification for ACC is still lacking. Hence, it is crucial to discover additional potent biomarkers for precise treatment and prognosis estimation in ACC. This study employed information from the TCGA and GEO databases to investigate the molecular classification, prognostic factors, immune infiltration, gene signature construction, and drug sensitivity of ACC from the perspective of IRGs and provided strong support for treatment decision-making and prognostic assessment of ACC patients.

Unsupervised clustering analysis categorized the patients into two subtypes by 2483 IRGs. Our findings revealed that the OS in patients with subtype 2 exhibited a notably greater value compared to subtype 1, suggesting that these IRGs may potentially impact the prognosis of ACC. To enhance the investigation of the involvement of these IRGs in ACC risk classification, we employed two subtypes as the primary characteristics. Through univariate Cox and LASSO regression analysis, we identified three genes (PRKCA, LTBP1, BIRC5). Subsequently, we developed an ACC IRG signature to shed light on the biological understanding of IRGS in ACC prognosis. The IRGS demonstrated its effectiveness in risk stratification across different cohorts. Furthermore, it demonstrated the ability to autonomously forecast the outcome of ACC by merging with additional clinical characteristics. It is noteworthy that individuals classified into the high-risk group experience a less favorable outcome compared to the low-risk category. To summarize, the signature is anticipated to improve our comprehension of the origin and progression of ACC, ultimately aiding in the advancement of precise clinical management approaches.

Based on the research, it was revealed that there are three IRGS genes that are implicated in signaling pathways associated with both the immune system and the formation of tumors. Prior research has shown that the initiation of PRKCA prompts subsequent tumor-promoting consequences in various forms of cancer. This happens via the MAPK/ERK [32] and PI3K/AKT signaling pathways [33]. These impacts include the suppression of cell death and the stimulation of cell growth, movement, infiltration, and blood vessel formation. Consequently, these processes collectively contribute to facilitating tumor progression. Regarding the prognosis, the presence of PRKCA in oral tongue squamous cell carcinomas (OTSCC) was linked to a lack of smoking and alcohol consumption history, leading to reduced OS [34]. PRKCA overexpression has been associated with reduced survival outcomes not only in ACC patients but also in lung adenocarcinoma [35]. The primary function of LTBP1 is to have a significant impact on the extracellular matrix and attach to the extracellular trap substances of the transforming growth factor-β (TGF-β) group, including TGF-β and BMPs (bone morphogenetic proteins), thereby controlling their activation and release [36]. Elevated LTBP1 levels in NK/T cell lymphoma (NKTCL) result in heightened TGFβ-1 secretion and release within the TME. Blocking the interaction between TGF-β and LTBP1 can impede the activity of natural killer/T cells by deactivating the TGF-β/Smad and p38MAPK signaling pathways, consequently impacting the advancement of lymphoma [37]. A previous investigation has presented proof that LTBP1 could function as a predictive risk element in instances of esophageal squamous cell carcinoma, and the elevated LTBP1 expression has been observed to display a favorable association with lymphatic metastasis in individuals suffering from esophageal squamous cell carcinoma [38]. In addition, Fu et al. reported that glioblastoma (GBM) cells showing increased LTBP1 expression demonstrate heightened abilities in terms of proliferation and migration when compared to cells with lower LTBP1 levels. Furthermore, this occurrence is associated with a more negative prognosis [39]. BIRC5 is widely acknowledged as a molecule that inhibits cell death and promotes cell proliferation and the advancement of tumors. Hence, it is widely recognized as a hopeful contender for therapeutic treatment [40]. Increased expression of BIRC5 in cancerous tissues has been demonstrated to enhance the formation of new blood vessels, stimulate cellular proliferation, and impede programmed cell death [41]. Multiple research studies have indicated that a significant number of cancer types frequently exhibit elevated levels of BIRC5, which is linked to resistance against chemicals and an unfavorable prognosis in individuals with cancer, such as lung adenocarcinoma [42], neuroblastoma [43], and glioma [44]. Furthermore, BIRC5 holds promise as a prospective candidate for identifying and predicting biomarkers for the identification, assessment, or prediction of breast cancer in patients [45].

In terms of signaling pathways, we noted that DEGs in the high-risk and low-risk groups were primarily enriched in immune regulation, inflammation, pathology, and biological processes associated with cell signal transduction. The GO enrichment results comprised pathways involving cytokine-mediated signaling, cell chemotaxis, receptor ligand activity, and signaling receptor activator activity, among others. The KEGG analysis comprises pathways such as cytokine-cytokine receptor interaction, viral protein interaction with cytokine and cytokine receptor, the chemokine signaling pathway, and neuroactive ligand-receptor interaction, among others. Our findings are also supported by molecular typing using IRGs in head and neck cancer [46]. To summarize, the findings of these enrichment analyses emphasize the contrasting functional characteristics of high-risk and low-risk groups, specifically in relation to immune responses, cell signaling, and potentially viral interactions. The Janus kinase-signal transducer and activator of transcription (JAK-STAT) pathway is activated by cytokine-mediated signaling, leading to changes in gene expression within cells. Consequently, this pathway regulates the activity, differentiation, and function of immune cells [47]. Various diseases, including AML/MDS, have been thoroughly investigated for immune system activation by extensively studying cytokines and their corresponding receptors. The potential to provide valuable insights into individual prognosis can be found in IL-6 and IL-10, as well as their corresponding receptors [48]. The presence of chemokines in tumors was linked to inflammation caused by T-cells [49]. In related studies, the neuroactive ligand-receptor interaction signaling pathway plays a direct and key role in the function of the nervous system [50].

Based on the attributes of IRGS, ACC patients were classified into different risk groups. The results showed that the group with low risk exhibited a higher levels of M2 macrophages, activated NK cells, and CD8^+^T cells. The results strongly indicate that there are different immune profiles between the high-risk and low-risk groups. Furthermore, our findings indicate a negative correlation between elevated immune score, stromal score, and microenvironment score and a low-risk score in patients. Similar findings were noted in patients with TME-associated subtypes of ACC [5]. In the same way, the risk score showed a strong correlation with elevated immune scores and the presence of tumor-infiltrating immune cells in melanoma. Additionally, the low-risk group exhibited the presence of CD8^+^T cells and NK cells [51]. Furthermore, based on the findings, we employed various techniques for immune infiltration to guarantee result stability. In the past few years, the rise of ICIs and targeted immunotherapy has become a crucial method of personalized treatment for patients with ACC. Recent studies indicate that the therapy of ICIs boosts the function of T cells by obstructing CTLA-4, PD-1, or PD-L1. There exists a direct relationship between the increased concentrations of ICIs and the efficacy of immunotherapy. This finding is of considerable importance in the context of prognosticating treatment responses [52]. Through an examination of the correlation between IRGS and ICIs manifestation, it was discovered that ten prevalent ICIs exhibited significant expression in the low-risk category, indicating heightened immune system activity within this group. In a highly inflammatory tumor setting, the increased expression of these ICIs could trigger a distinct feedback mechanism, fostering a dynamic immune environment that has the potential to enhance patient outcomes [53]. The high expression of ICIs may be linked to a better survival outcome in the low-risk group, considering our findings. Furthermore, apart from utilizing these typical indicators, we have also chosen several potential immunotherapy predictors to evaluate the effectiveness of our signature in forecasting the reaction of ACC patients to ICIs. We discovered that individuals in the low-risk group exhibited elevated IPS, GEP, and MAIS scores, with all three scores displaying an inverse relationship with the risk scores. The results indicate that there are notable variations in the immune microenvironment between high-risk and low-risk groups, with the low-risk group having a higher probability of gaining advantages from immunotherapy [54].

The efficacy of ICIs in different types of tumors, such as non-small-cell lung cancer (NSCLC), metastatic gastrointestinal cancer, and colorectal cancer, has been assessed using TMB as a potential biomarker [55,56,57,58]. A higher TMB results in an increased number of novel antigens, enhancing the likelihood of T cell detection and improving the clinical response to ICIs [59]. In line with prior research on ACC, it was observed that individuals with high TMB experienced poorer survival results and exhibited a negative association with their risk score. Conversely, those in the low TMB category demonstrated favorable survival outcomes. Notably, these findings align with the outcomes of diffuse glioma and neuroblastoma, yet contrast with the findings of NSCLC, suggesting that TMB displays distinct prognostic traits across various cancer types [55,60]. Hence, a comprehensive examination of pan-cancer is necessary to completely understand its function. Furthermore, following risk stratification, the TMB status had no impact on the prediction of IRGS. The prognosis of the low-risk group has been better consistently, suggesting that IRGS possessed superior prognostic prediction capability, aligning with the findings of low-grade glioma [61]. We additionally investigated the association between IRGS and the effectiveness of antineoplastic medications for the created IRGS. Three frequently employed chemotherapeutic medications were chosen for the treatment of ACC. It was observed that the IC50 of cisplatin, gemcitabin, and docetaxel in the low-risk group was higher than that in the high-risk group, suggesting that high-risk patients may derive greater benefits from these commonly prescribed medications. Based on the current treatment guidelines, it is recommended to use etoposide, docetaxel, and cisplatin in combination with oral mitotane (EDP-M) as the primary treatment for metastatic ACC [62]. However, research has indicated that the combination of cisplatin and gemcitabin is also effective in treating ACC [63].

The current investigation extensively examined the possible influence of immune-associated genes on the prognosis of ACC individuals, unveiling their potential in terms of prognosis, immune attributes, and immunotherapy from various angles. Our main emphasis was on three IRGs, and we projected the OS of ACC patients by creating feature labels using these genes. Furthermore, the validation of the signature’s predictive capability is reaffirmed simultaneously. Of course, some limitations should be noted in this study. To begin with, ACC is an uncommon and diverse form of cancer, making it challenging to acquire extensive patient samples [64]. Nevertheless, the gene signature we have built can also be validated in two separate external validation sets. Secondly, this is a retrospective study, and it is inevitable that there may be selection bias and low statistical ability.

## 4. Materials and Methods

### 4.1. Data Acquisition and Pre-Processing

#### 4.1.1. Data Download

In this work, we obtained RNA-seq data and the most recent clinical data of ACC patients from The Cancer Genome Atlas (TCGA) (https://xenabrowser.net/, accessed on 1 June 2023), and a grand total of 79 RNA-seq data samples were acquired. Key clinical features comprise age, gender, clinical stage, T stage, M stage, and N stage (Appendix A). Furthermore, the validation datasets GSE33371 and GSE10927 were obtained from the GEO (https://www.ncbi.nlm.nih.gov/geo/, accessed on 2 June 2023) repository. According to the screen criteria, GSE33371 contains 23 samples with clinical features, and GSE10927 contains 24 samples with clinical features. Appendix A displays the detailed clinicopathological characteristics of patients with ACC. Subsequently, we downloaded the data of 2483 IRGs from the ImmPort (https://immport.niaid.nih.gov, accessed on 23 July 2023) for analysis.

#### 4.1.2. Data Pre-Processing

For this investigation, gene expression profile data were measured on the Illumina platform. We marked these genes utilizing the annotation document named “gencode.v22.annotation.gene.probeMap”. The measurement of gene expression profiles was done through the estimation of transcripts per million (TPM) and fpkm values converted to a log2 scale. Differential expression analysis utilized data in count format.

#### 4.1.3. Data Screen Criteria

First, excluded were samples that did not have clinical follow-up information and those with a survival period shorter than 30 days in TCGA-ACC patients. Second, the gene expression data for all samples were complete. Finally, normal tissue samples were not considered, and only tumor samples were included. Figure 9 displays the workflow chart.

### 4.2. Molecular Subtype Analysis

After intersecting the IRGs with the gene expression profile in TCGA-ACC, we used the “ConsensusClusterPlus” package [65] for unsupervised clustering and identified molecular subtypes. Then, we applied the “pam” algorithm and Pearson’s method for similarity measurement in hierarchical clustering. Sampling 80% of the data was done in each repetition of the clustering process, which was repeated 1000 times. The ideal value for the clustering number k was determined by analyzing the CDF, Delta area plot, and PAC algorithm [66], and only the stable clustering results were chosen. 

### 4.3. Differential Expression Analysis

In this study, we employed the “DESeq2” package [67] to examine the variation in expression among various subtypes. The significant DEGs were identified by applying screening criteria of |log2FC| > 1 and *p* < 0.05. These DEGs were then compared with the IRGs. Afterward, the volcano map is employed for visualization. 

### 4.4. Development and Verification of IRGs Signature 

Initially, we conducted univariate Cox regression analysis on the potential IRGs using the “survival” R package, taking into account genes with *p*-values below 0.05 as statistically noteworthy. Afterwards, we employed LASSO regression analysis to further identify prognostic-associated genes and obtained noteworthy outcomes from univariate Cox regression [68]. The lambda values were determined using a 10-fold cross-validation approach. Here, lambda = 0.20 with an appropriate partial likelihood residual was selected. The risk score of each sample was obtained by the following formula: Riskscore = Expression_IRGS1_ × Coefficient_IRGS1_ + Expression_IRGS_2 × Coefficient_IRGS2_ + … + Expression_IRGSn_ × Coefficient_IRGSn_ [69]. The risk assessment of TCGA-ACC cancer patients resulted in the categorization of individuals into high- and low-risk groups based on the median risk score. Ultimately, the identical approach is employed to authenticate the outcomes in the validation set. To ascertain the independence of IRGS as a predictive marker, univariate Cox analysis was conducted to assess the risk score in conjunction with relevant clinical variables.

### 4.5. Analysis of Immune Microenvironment Characteristics

In our study, the “CIBERSORT” algorithm was used to calculate 22 representative immune cells based on the LM22 profile and normalized expression matrix [70]. Simultaneously, we employed the “ESTIMATE” algorithm to quantify the immune microenvironment of ACC patients. To ensure the robustness and consistency of our findings, we also integrated the xCell algorithm into our analysis, enabling the evaluation of the infiltration ratios of 64 specific immune cell types and facilitating the comparative assessment of their levels of infiltration among distinct risk groups.

### 4.6. Analysis of Somatic Mutations

To determine the mutational burden of ACC, we employed the R package “TCGAbiolinks” [71] to fetch ACC mutation data. Then, we used the “maftools” package [72] to analyze the mutation data and obtain TMB for each patient. Visualizing the results with a waterfall diagram. Based on the median, TMB was categorized into two groups: high-TMB and low-TMB, and the mutation characteristics were subsequently compared between the different risk groups. Additionally, the KM curve was utilized to examine the relationship between TMB, different risk levels of TMB, and the prognosis of ACC.

### 4.7. Biological Functional Analysis

To gain insight into the possible molecular mechanisms underlying the DEGs in IRG subtypes and risk groups, the “clusterprofiler” package was used to conduct “GO” enrichment and “KEGG” pathway analyses [73]. The differential functions and pathways were selected via the “BH” method with an adjusted *p* value < 0.05. A bubble diagram was used to depict the major functions and pathways. Furthermore, we used GSVA to directly access the downloaded C2 reference gene sets, which consist of KEGG data and C7 immunologic signature gene sets. The pathway difference was analyzed using the “limma” package, and the analysis results were visualized through a heatmap.

### 4.8. Predictive Analysis of Immunotherapy 

In order to enhance our comprehension of the correlation between IRGS and specific prognostic immunotherapy indicators, we performed a comparative examination between different risk groups. The immunogenicity of tumors can be evaluated, and the effectiveness of different types of ICIs therapies can be predicted using IPS. The evaluation of four different immunophenotypes, namely antigen presentation, effector cells, inhibitory cells, and checkpoints, is encompassed by IPS. The immunogenicity of the IPS was assessed using a z-score ranging from 0 to 10, where a higher value suggests an increased immunogenic response. Simultaneously, we employed GEP [74] and MIAS (Microenvironment, Immune Response, and Anti-tumor Sensitivity) scores as indicators for forecasting patients’ reactions to immunotherapy.

### 4.9. Drug Sensitivity Analysis 

The presence of diversity in ACC leads to significant differences in how patients respond to the same pharmacological treatments. We utilized the Genomics of Drug Sensitivity in Cancer (GDSC) database as our main training dataset to examine the variation in the responsiveness of anti-cancer medications among patients belonging to diverse risk categories. Afterwards, we utilized the R “oncoPredict” package to determine the IC50 values of three traditional chemotherapy drugs used in the clinical treatment of ACC patients [75]. It should be emphasized that cisplatin is an ACC treatment drug that has been extensively researched. In addition, in vitro studies have also confirmed the effectiveness of taxanes [76]. Currently, several investigations have conducted research on the utilization of cisplatin in conjunction with gemcitabine and docetaxel for treating individuals diagnosed with advanced adrenocortical carcinoma.

## 5. Conclusions

We identified two new subtypes of ACC based on IRGs, constructed a prognostic gene signature for IRG-related risks, including PRKCA, LTBP1, and BIRC5, and verified its predictive ability. The IRGS could potentially be a dependable predictive biomarker for individualized care of ACC patients, offering a theoretical foundation for accurate immunotherapy in ACC patients.

## Figures and Tables

**Figure 1 ijms-24-15465-f001:**
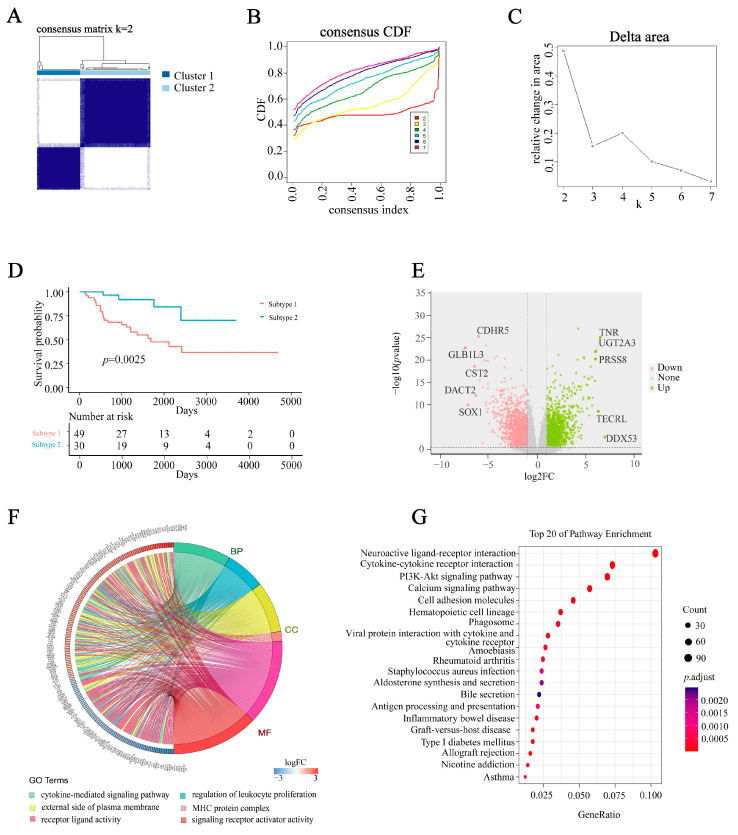
Characterization of immune properties and biological roles of molecular subtypes. (**A**–**C**) Identification of two molecular subtypes. (**D**) Kaplan–Meier survival analysis of OS between molecular subtypes. (**E**) The volcano plot of DEGs between two subtypes. (**F**) GO analysis showing the pathways enriched in DEGs. (**G**) KEGG analysis showing the pathways enriched in DEGs.

**Figure 2 ijms-24-15465-f002:**
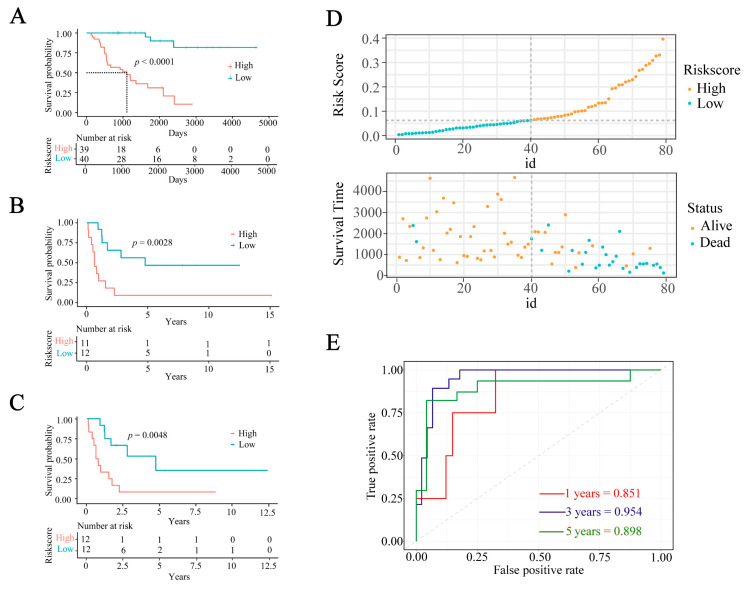
The prognostic characteristics of IRGS. (**A**) The Kaplan–Meier analysis of TCGA-ACC patients. (**B**) The Kaplan–Meier analysis of the GSE33371 dataset. (**C**) The Kaplan-Meier analysis of the GSE10927 dataset. (**D**) Risk score and survival time distribution in high- and low- risk groups. (**E**) The receiver operating characteristic curve in the TCGA-ACC.

**Figure 3 ijms-24-15465-f003:**
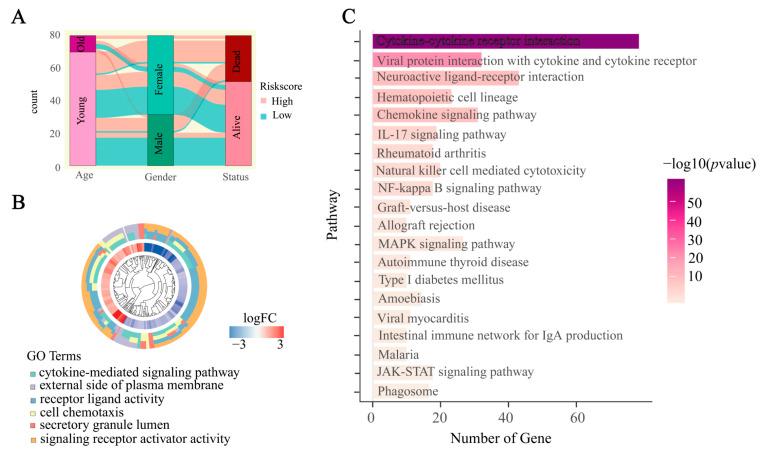
Functional analysis of IRGS. (**A**) The distribution of clinical features of TCGA-ACC samples. (**B**) The circular cluster diagram showing the results of GO pathway enrichment analysis. (**C**) The bar chart showing the top 20 most significant pathways in the KEGG pathway enrichment analysis.

**Figure 4 ijms-24-15465-f004:**
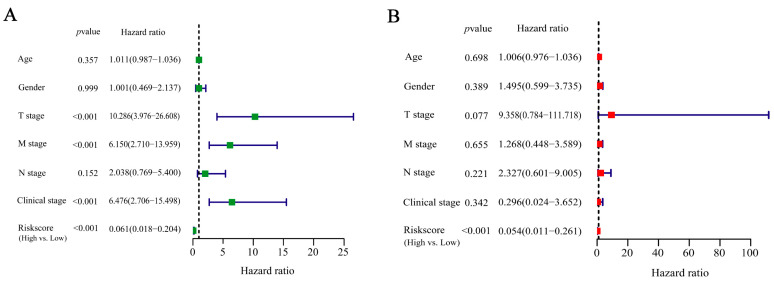
Univariate and multivariate Cox analysis results in TCGA-ACC patients. (**A**) The forest plot illustrating the results of the univariate Cox regression analysis in the TCGA-ACC patients. Green color blocks show the results of univariate Cox analysis. (**B**) The forest plot illustrating the findings of multivariate Cox regression analysis in the TCGA-ACC patients. Red color blocks show the results of multivariate Cox analysis. (*p* < 0.05 was considered significant).

**Figure 5 ijms-24-15465-f005:**
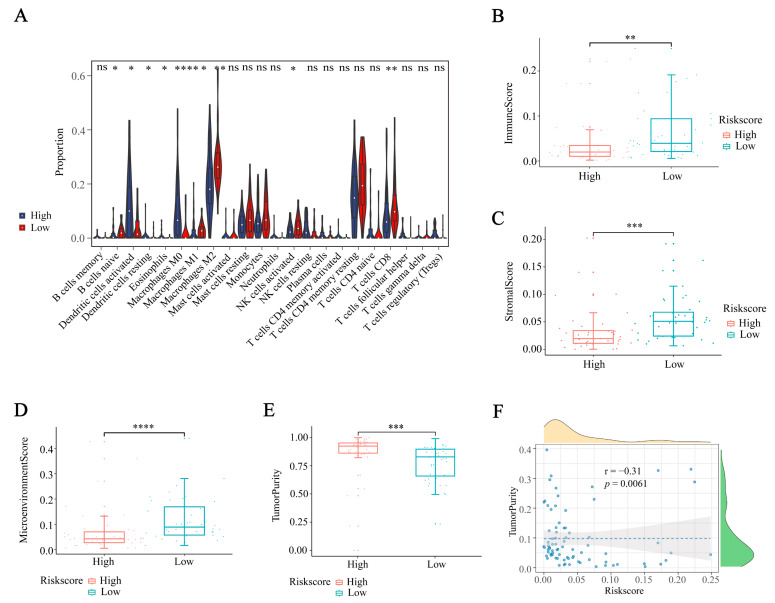
Exploring the landscape of immune cell infiltration in IRGS (**A**) CIBERSORT algorithm showing 22 immune cell infiltration levels. (**B**–**D**) xCell algorithm evaluating the infiltration levels of 64 immune cell types. (**E**,**F**) ESTIMATE algorithm showing the relationship between tumor purity and risk score. Data in (**A**–**E**) were analyzed by the Wilcoxon test; (ns, no significance, * *p* < 0.05, ** *p* < 0.01, *** *p* < 0.001, and **** *p* < 0.0001).

**Figure 6 ijms-24-15465-f006:**
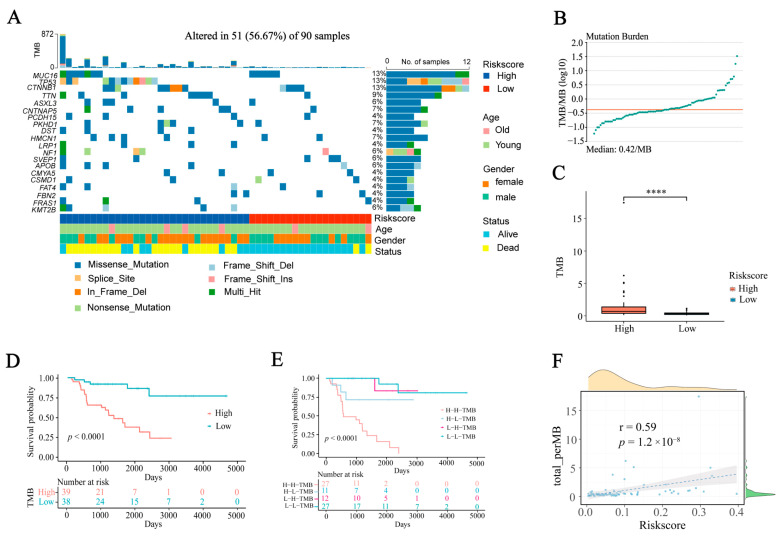
The correlation between IRGS and TMB. (**A**) Waterfall plot illustrating the top 20 highly mutated genes of TCGA-ACC. (**B**) Median value of TMB distribution in TCGA-ACC patients. (**C**) Comparison of TMB scores in different risk groups. (**D**) Kaplan-Meier curves of OS for high- and low- TMB groups. (**E**) KM curves of OS for ACC stratified by both risk score and TMB. (**F**) Correlation analysis between risk scores and TMB. Data in (**C**) were analyzed by the Wilcoxon test; (**** *p* < 0.0001).

**Figure 7 ijms-24-15465-f007:**
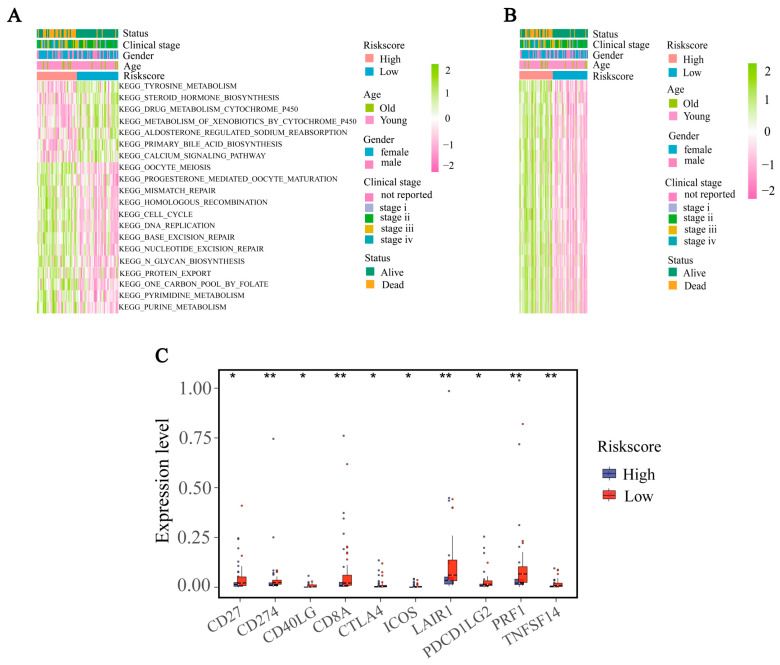
KEGG and immunologic pathways in different risk groups and clinical characteristics. (**A**) The distribution of the top 20 most significant KEGG pathways in different risk groups. (**B**) Heatmap showing the immunologic pathways in different risk groups. (**C**) The box plot illustrating the expression levels of ten immune checkpoints in different risk groups. Data in (**C**) were analyzed by the Wilcoxon test; (* *p* < 0.05, ** *p* < 0.01).

**Figure 8 ijms-24-15465-f008:**
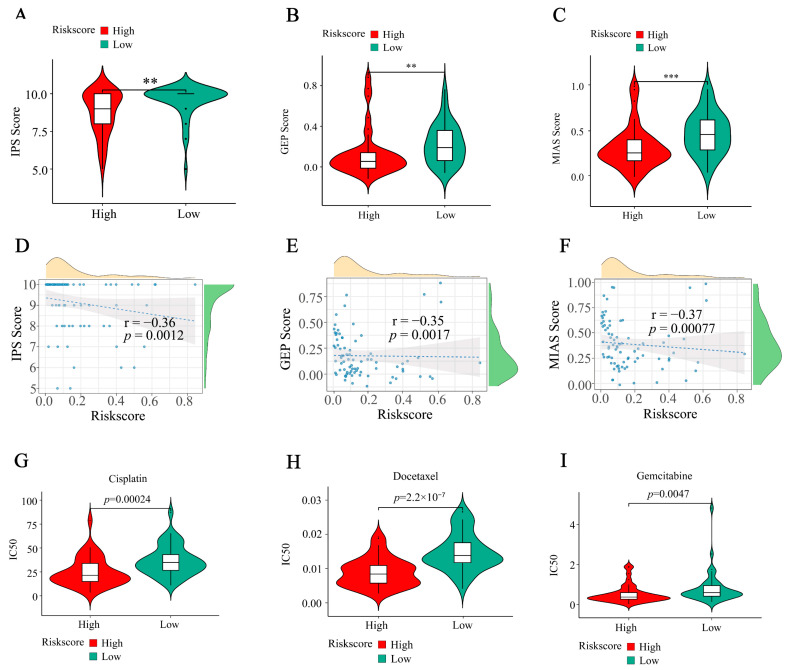
The correlation between IRGS and immunotherapy indicators. (**A**–**C**) Violin plot of three ICI-related indicators between the high-risk group and the low-risk group in the TCGA-ACC dataset. (**D**–**F**) The relationship between three ICI-related indicators (IPS, GEP, and MAIS) and risk scores, respectively. (**G**–**I**) Comparison of three chemotherapeutic drugs between high- and low-risk groups in the TCGA-ACC dataset. Data in (**A**–**C**,**G**–**I**) were analyzed by the Wilcoxon test; (** *p* < 0.01, *** *p* < 0.001).

**Figure 9 ijms-24-15465-f009:**
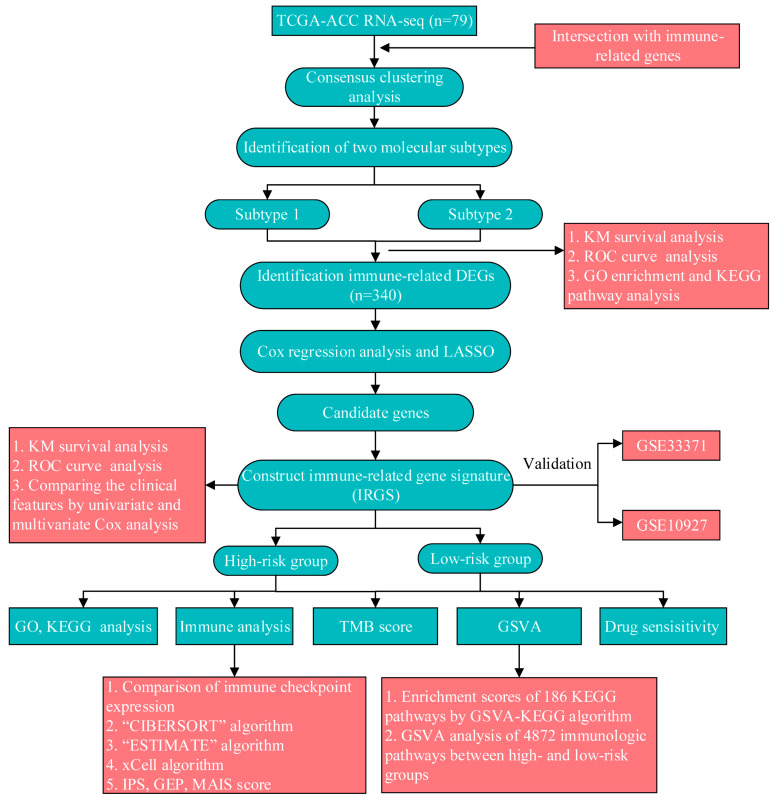
The workflow chart in this study.

## Data Availability

The corresponding data and results were generated by the TCGA (https://xenabrowser.net/, accessed on 1 June 2023), GEO (http://www.ncbi.nlm.nih.gov/geo/, accessed on 2 June 2023), and ImmPort databases (https://immport.niaid.nih.gov, accessed on 23 July 2023).

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
