# Peer review of "Identification of Molecular Subtypes and Prognostic Characteristics of Adrenocortical Carcinoma Based on Unsupervised Clustering"

_ijms, 2023, doi:10.3390/ijms242015465_

Round 1
Reviewer 1 Report
Dear Authors,
I have found you manuscript as an interesting and topical in the field. However, there are some minor things I would like ask you ti change: 1) Materials and methods. Could you, please, ad some subsections 4.1.1, 4.1.2., to develop more clear the data analysis steps. Also clarify please exclusion/inclusion criteria and place then in these subsections because it is not very easy to follow to your protocol now... The Fig. 9 is OK, but I think that the description of the protocol should be more clarified!
2) Conclusions. Please, avoid some extra phrases, like "In summary,..."from this part, as they are completely useless.
Author Response
Comments 1: Materials and methods. Could you, please, add some subsections 4.1.1, 4.1.2., to develop more clear the data analysis steps. Also clarify please exclusion/inclusion criteria and place then in these subsections because it is not very easy to follow to your protocol now. The Fig. 9 is OK, but I think that the description of the protocol should be more clarified!.
Response 1: Thanks for your suggestion. We agree this suggestion and have revised the data analysis steps and protocol content in the manuscript. We have modified the data acquisition and download section of the article, detailed its steps, and clarified the screening criteria for samples to develop clearer data analysis steps. They can be found in page 14 lines 434-441, page 15 lines 442-470. Furthermore, we have also optimized the flow chart, as shown in Figure 9.
Comments 2: Conclusions. Please, avoid some extra phrases, like “In summary,...”from this part, as they are completely useless.
Response 2: Thanks for your comments, we have corrected the expression of the conclusions appropriate in page 19 lines 553-554.
Please see attachment for details.

Reviewer 2 Report
In this paper, the authors present an interesting study on their effort to examine the connection between IRGs and ACC immunotherapy using an unsupervised clustering method aiming to add more data in the limited field of ACC and investigate the potential impact of IRGs on the prognosis of ACC.
The introduction is very well constructed providing efficiently the background and novelty information based on the international current literature. Moreover, Materials and Methods are described thoroughly and this gives the study the advantage of being reproducible by other investigators. The results are well presented and the reported hypothesis as conclusion that IRGS could potentially be a dependable predictive biomarker for individualized care of ACC patients, offering a theoretical foundation for accurate immunotherapy in ACC patients is very promising.
English language is quite fine.
Author Response
Comments 1: In this paper, the authors present an interesting study on their effort to examine the connection between IRGs and ACC immunotherapy using an unsupervised clustering method aiming to add more data in the limited field of ACC and investigate the potential impact of IRGs on the prognosis of ACC. The introduction is very well constructed providing efficiently the background and novelty information based on the international current literature. Moreover, Materials and Methods are described thoroughly and this gives the study the advantage of being reproducible by other investigators. The results are well presented and the reported hypothesis as conclusion that IRGS could potentially be a dependable predictive biomarker for individualized care of ACC patients, offering a theoretical foundation for accurate immunotherapy in ACC patients is very promising.
Response 1: We are grateful to you for reviewing our paper and your positive feedback.